# Syndiotactic Poly(4-methyl-1-pentene)-Based Stereoregular Diblock Copolymers: Synthesis and Self-Assembly Studies

**DOI:** 10.3390/polym14224815

**Published:** 2022-11-09

**Authors:** Yu-Chuan Sung, Pei-Sun Huang, Shih-Hung Huang, Yeo-Wan Chiang, Jing-Cherng Tsai

**Affiliations:** 1Department of Chemical Engineering, National Chung Cheng University, Chiayi 62142, Taiwan; 2Department of Materials and Optoelectronic Science, National Sun Yat-sen University, Kaohsiung 80424, Taiwan

**Keywords:** stereoregularity, end-functionalization, block copolymer, metallocene, ATRP, self-assembly

## Abstract

Syndiotactic poly(4-methyl-1-pentene) (sP4M1P)-based stereoregular diblock copolymers, namely sP4M1P-*b*-polystyrene and sP4M1P-*b*-polymethylmethacrylate, were prepared from an α-bromoester-capped sP4M1P macroinitiator, which was chain extended with styrene and methyl methacrylate, respectively, via the atom transfer radical polymerization reaction. The α-bromoester-capped sP4M1P was generated by the esterification of hydroxyl-capped sP4M1P with α-bromoisobutyryl bromide. The hydroxyl-capped sP4M1P was synthesized by inducing a selective chain transfer reaction to aluminum during the syndiospecific polymerization of 4-methyl-1-pentene in the presence of a syndiospecific metallocene catalyst. As stereoregular diblock copolymers are difficult to prepare using existing methods, the current study offers an effective process for the preparation of sP4M1P-based stereoregular diblock copolymers. These copolymers were found to have well-defined architectures and they can undergo molecular self-assembly into ordered nanostructures, as evidenced by small-angle X-ray scattering analyses.

## 1. Introduction

Block copolymers [1,2,3] are generally composed of two or more different polymer blocks that are joined with covalent bonds. Because of their unique chemical architecture, block copolymers can self-organize into ordered nanopatterns, which may be suitable as nanomaterial templates [4,5,6,7,8]. Block copolymers are typically prepared via living polymerization reactions, as these synthetic processes offer convenient methods for the construction of block copolymers through the sequential polymerization of various monomers [9,10,11]. Despite this, the structure of α-olefin monomers that can be polymerized through living polymerization is limited. Furthermore, most living polymerization reactions do not facilitate stereoregularity control for the polymerization of α-olefins. As a result, the synthesis of α-olefin-based block copolymers, which have a broad variety of chemical architectures (e.g., α-olefin bearing bulky neighboring groups) and a stereoregular skeleton (e.g., in terms of tacticity [12,13] or chirality [14,15]), remains a challenge. 

In this paper, we aim to synthesize structurally well-defined syndiotactic poly(4-methyl-1-pentene) (sP4M1P)-based stereoregular diblock copolymers, which are difficult to prepare using the common living polymerization methods. Of note, a polymer of 4-methyl-1-pentene (4M1P) has been industrially polymerized using the Ziegler-Natta catalyst. The resulting isotactic poly(4-methyl-1-pentene) acts as a thermoplastic, with applications in gas-permeable packaging as well as medical and laboratory equipment. Block copolymers of poly(4-methyl-1-pentene) were prepared by Wu [16] and by Ricci [17], respectively, through living polymerization using a nickel diamine catalyst. As the nickel diamine catalysts used in their studies do not facilitate stereoregularity control, poly(4-methyl-1-pentene)-based block copolymers prepared via these synthetic processes lack stereoregular configuration in the polymer microstructure. Recently, Sita reported the syntheses of stereoregular isotactic and syndiotactic block copolymers of poly(4-methyl-1-pentene) using living Ziegler-Natta catalysts [18,19]. In the present study, a combination of a metallocene catalyst [20] and atom transfer radical polymerization [21] is used to prepare syndiotactic poly(4-methyl-1-pentene) (sP4M1P)-based stereoregular block copolymers with tacticity control and precise linkage between polymer blocks. In our prior studies, we demonstrated that the self-assembly of stereoregular block polymers typically leads to the generation of ordered nanopatterns with unusual nanomorphologies [22,23,24]. As the 4M1P monomer contains a bulky isobutyl neighboring group, the self-assembly of sP4M1P-based stereoregular diblock copolymers would allow the investigation of the unique steric interaction between these bulky substituents at the molecular level.

## 2. Experimental Section

### 2.1. General Procedure

All reactions and manipulations were conducted under a nitrogen atmosphere using the standard Schlenk line or dry box techniques. Solvents and common reagents were commercially obtained and used either as received or purified by distillation with sodium/benzophenone. Styrene (99%), 4M1P (98%), and methyl methacrylate (99%) purchased from Aldrich were dried over calcium hydride and distilled under vacuum before use. Me_2_C(Cp)(Flu)ZrCl_2_ was synthesized using a method described in the literature [25,26]. Triethylaluminum (TEA, 1 M in hexane) and trimethylaluminum (TMA, 1 M in hexane) were purchased from Aldrich and used as received. Methylaluminoxane (MAO, 14% in toluene), purchased from Albemarle, was dried under vacuum to remove residual TMA. The resulting TMA-free MAO was diluted in toluene to the desired concentration before use [27]. Copper bromide (98%), α-bromoisobutyryl bromide (98%), 1,10-phenanthroline (>99%), and 1,1,4,7,10,10-hexamethyltriethylenetetramine (>99%) were purchased from Aldrich and used as received. Oxygen (purity > 99%) was obtained from Matheson and used as received.

### 2.2. Preparation of OH-Capped sP4M1P

Representative experiment (for entry 6 in Table 1): A 150 mL stainless steel reactor, equipped with a magnetic stirrer, was allowed to dry at 80 °C under vacuum. After being refilled with nitrogen, the reactor was maintained at 20 °C and then charged sequentially with 30 mL of toluene, 5.0 mmol of MAO, and 2.5 μmol of Me_2_C(Cp)(Flu)ZrCl_2_. After the reactor was allowed to stir at 20 °C for 5 min, it was charged with 2.0 mmol of TEA and then 2 mL of 4M1P to initiate the polymerization reaction. Polymerization was conducted at 20 °C for 24 h, after which the polymer solution was treated with oxygen at a flow rate of 12 mL/min for 1 h. The solution was then charged with H_2_O_2_ (4 mL, 30% in H_2_O). After the solution was stirred at room temperature for 30 min, it was treated with excess methanol (ca. 40 mL), which led to the deposition of the OH-capped sP4M1P as a white precipitate. The resulting polymer was isolated after filtration and dried under vacuum to produce 0.53 g of OH-capped sP4M1P. *M*_n_ = 9890 and *M*_w_/*M*_n_ = 1.65, as determined by gel permeation chromatography (GPC) in tetrahydrofuran (THF) at 45 °C.

### 2.3. Fractionation of OH-Capped sP4M1P 

Exactly 0.53 g of the OH-capped sP4M1P sample (*M*_n_ = 9890, *M*_w_/*M*_n_ = 1.65, entry 8 in Table 1) was placed in a Soxhlet extractor and allowed to undergo extraction with boiling methyl ethyl ketone (MEK, 150 mL) for 24 h. The MEK-insoluble polymer was then collected and allowed to undergo a second Soxhlet extraction with boiling THF/MEK (1:1, 150 mL) for 24 h. The collected THF/MEK solution was concentrated to 50 mL under vacuum. The resulting solution was then charged with 50 mL of methanol. This led to the deposition of the boiling THF/MEK soluble polymer as a white precipitate. The polymer was isolated by filtration and dried under vacuum to produce 0.20 g of OH-capped sP4M1P [*M*_n_ = 10,500 and *M*_w_/*M*_n_ = 1.40 (sample 3 in Table 2), as determined by GPC in THF at 45 °C]. The boiling THF/MEK-insoluble polymer was collected and dried under vacuum to produce 0.21 g of OH-capped sP4M1P [*M*_n_ = 14,500 and *M*_w_/*M*_n_ = 1.32 (sample 1 in Table 2), as determined by GPC in THF at 45 °C].

### 2.4. Preparation of α-Bromoisobutylester-Capped sP4M1P 

In a 200 mL round bottom flask, 0.21 g of the fractionated OH-capped sP4M1P (*M*_n_ = 14,500 and *M*_w_/*M*_n_ = 1.32; sample 1 in Table 2) was charged sequentially with 50 mL of toluene, 1.0 mL of triethylamine, and 0.5mL of 2-bromoisobutyryl bromide (in excess). The resulting solution was heated to 60 °C to conduct the esterification reaction at 60 °C for 4 h. The reaction solution was then cooled to room temperature and charged with excess methanol (50 mL), which led to the deposition of the sP4M1P-based polymer as a pale brown precipitate. The resulting product of the reaction was collected through filtration, washed with methanol, and dried under high vacuum to produce 0.24 g of α-bromoisobutylester-capped sP4M1P (*M*_n_ = 14,500 and *M*_w_/*M*_n_ = 1.32, as determined by GPC in THF at 45 °C).

In another 200 mL round bottom flask, 0.28 g of the fractionated OH-capped sP4M1P (*M*_n_ = 10,500, *M*_w_/*M*_n_ = 1.40, sample 3 in Table 2) was allowed to undergo a similar esterification reaction (*vide supra*). This led to the production of 0.26 g of α-bromoisobutylester-capped sP4M1P (*M*_n_ = 10,500 and *M*_w_/*M*_n_ = 1.40, as determined by GPC in THF at 45 °C). 

### 2.5. Preparation of sP4M1P-b-PMMA

In a dry box, a 100 mL Schlenk flask equipped with a magnetic stirrer was charged sequentially with 0.20 g (0.014 mmol) of α-bromoisobutylester-capped sP4M1P (*M*_n_ = 14,500, *M*_w_/*M*_n_ = 1.32), 30 mL of toluene, 5.5 μmol of copper bromide, and 11.0 μmol of 1,10-phenanthroline. The Schlenk flask was removed from the dry box and heated to 95 °C in an oil bath for 30 min for the generation of the macroinitiator. The reaction vessel was then charged with 0.45 g of MMA (4.5 mmol) to initiate the chain extension reaction of the macroinitiator with MMA at 95 °C for 24 h. The polymer solution was then filtered through aluminum oxide to remove copper complexes. The resulting filtrate was charged with excess methanol (ca. 20 mL), which led to the deposition of the reaction product as an off-white precipitate. The resulting precipitate was isolated by filtration and dried under vacuum to produce 0.43 g of sP4M1P-*b*-PMMA [*M*_n_ = 32,000 and *M*_w_/*M*_n_ = 1.20 (sample 2 in Table 2), as determined by GPC in THF at 45 °C].

### 2.6. Preparation of sP4M1P-b-aPS

In a dry box, a 100 mL Schlenk flask equipped with a magnetic stirrer was charged sequentially with 0.20 g (0.019 mmol) of α-bromoisobutylester-capped sP4M1P (*M*_n_ = 10,500, *M*_w_/*M*_n_ = 1.40), 30 mL of toluene, 5.5 μmol of copper bromide, and 11.0 μmol of 1,1,4,7,10,10-hexamethyltriethylenetetramine. The reaction vessel was heated to 95 °C in an oil bath for 30 min for the generation of the macroinitiator. Then, the reaction vessel was charged with 0.35 g of styrene (3.4 mmol) to initiate the chain extension reaction of the macroinitiator with styrene at 95 °C for 24 h. The polymer solution was then filtered through aluminum oxide to remove copper complexes. The resulting filtrate was charged with methanol (ca. 20 mL), which led to the deposition of the reaction product as an off-white precipitate. The resulting precipitate was isolated by filtration and dried under vacuum to produce 0.36 g of sP4M1P-*b*-aPS [*M*_n_ =19,000 and *M*_w_/*M*_n_ = 1.34 (sample 4 in Table 2), as determined by GPC in THF at 45 °C].

### 2.7. Polymer Analysis

The molecular weight and molecular weight distribution (*M*_w_/*M*_n_) were determined through high-temperature gel permeation chromatography (Waters 150-CALAC/GPC) with a refractive index detector and a set of U-Styragel HT columns with pore sizes of 10^6^, 10^5^, 10^4^, and 10^3^ in series. The measurements were taken at 45 °C using tetrahydrofuran as the solvent. PS samples with narrow MWDs were used as the standards for calibration. The standards were in the range of the absolute molecular weight, namely 980 to 2,110,000; and the *R* squared value of the ideal calibrated line was limited to 0.999.

The ^1^H (500 MHz) and ^13^C NMR (125 MHz) spectra were recorded on a Bruker AV-500 NMR spectrometer. Polymer samples were dissolved into CDCl_3_ as the solvent. The recorded temperature was 60 °C.

### 2.8. Bulk Sample Preparation

All the bulk samples were prepared by dissolving the diblock copolymers in tetrahydrofuran (THF) at 60 °C. The concentration of the polymers in the solution was 5 wt%. The samples were subsequently cast onto Petri dishes, followed by a slow evaporation of the solvent at 30 °C for two days. The samples were further dried under vacuum at 60 °C for 2 h and then at room temperature for four days to completely remove the solvent. The resultant films of the block copolymers were studied using small-angle X-ray scattering (SAXS) and differential scanning calorimetry (DSC).

### 2.9. Small-Angle X-ray Scattering Measurement

Morphologies of the BCPs in the melt and crystalline state were characterized by small-angle X-ray scattering (SAXS). SAXS experiments were conducted at the synchrotron X-ray beamline 23A at the National Synchrotron Research Center (NSRRC) in Taiwan. The wavelength λ of the X-ray beam was 0.082657 nm. A pixel detector, Pilatus-1MF, was utilized for collecting the two-dimensional (2D) SAXS patterns. One-dimensional (1D) SAXS profiles were obtained by integration of the 2D SAXS patterns. The scattering angle of the SAXS pattern was calibrated using silver behenate with the first-order scattering vector *q** = 1.076 nm^−1^ (*q** = 4πλ^−1^ sinθ, where 2θ is the scattering angle).

## 3. Results and Discussion

### 3.1. Preparation of Hydroxyl-Capped sP4M1P by Selective Chain Transfer to Alkylaluminum

Chain transfer to aluminum has been reported to offer the end functionalization of various olefin polymers and generate a single chain transfer reaction product (e.g., a hydroxy-capped polyolefin polymer) by metallocene-catalyst-mediated polymerization reactions [28,29,30,31]. However, the end functionalization of 4M1P, which contains a bulky isobutyl neighboring group, has not been achieved. It could be quite challenging, as the bulky neighboring group may significantly reduce the polymerization rate, change the olefin insertion pattern, and alter the chain transfer route. In our efforts to synthesize the hydroxy-capped sP4M1P, a syndiospecific Me_2_C(Cp)(Flu)ZrCl_2_ complex was used as the catalyst to combine with two different alkylaluminum compounds (i.e., trimethylaluminum and triethylaluminum) which act as chain transfer agents. After polymerization, the polymer solution was in situ treated with O_2_/H_2_O_2_ to convert the air-sensitive aluminum end group into a stable hydroxyl end group. Thus, possible chain transfer pathways involved in the metallocene-catalyst-mediated polymerization can be identified by chain-end analyses of the sP4M1P samples through ^1^H and ^13^C-NMR analyses. The results of the polymerization studies are summarized in Table 1. 

As shown by entries 1–5 in Table 1, the syndiospecific polymerization of 4M1P using Me_2_C(Cp)(Flu)ZrCl_2_ as the catalyst and TMA as the chain transfer agent typically produced sP4M1P with two types of functional end groups, namely hydroxy and vinylene end groups. The formation of the hydroxy end group can be rationalized by the 2,1-insertion of a terminal 4M1P monomer unit which underwent chain transfer to aluminum to produce the preliminary aluminum-capped sP4M1P. Subsequently, in situ oxidation of the aluminum end group with ozone and hydrogen peroxide produced the hydroxy-capped sP4M1P as the major chain transfer reaction product. By contrast, the vinylene end group was generated by the 2,1-insertion of a terminal 4M1P unit which underwent β-hydride elimination chain transfer to produce the vinylene-capped sP4M1P. The detailed chain transfer pathways for the production of hydroxy and vinylene end-capped sP4M1P are illustrated in Figure 1.

Efforts to suppress the β-hydride elimination transfer by reducing polymerization temperature and increasing the concentration of TMA did not result in the production of pure OH-capped sP4M1P despite these reaction conditions enhancing the amount of OH-capped sP4M1P generated in the polymer solution (see entries 1–5 in Table 1). 

As TEA was reported to be a more efficient chain transfer agent compared with TMA, the preparation of OH-capped sP4M1P using TEA in place of TMA as the chain transfer agent was investigated. Entries 6 and 7 in Table 1 show that at higher temperature (>40 °C), the syndiospecific polymerization of 4M1P did not lead to the generation of OH-capped sP4M1P as the only polymerization product. However, using TEA as the chain transfer agent and mediating the polymerization at a lower temperature (<20 °C) generated pure OH-capped sP4M1P as the only chain transfer reaction product (see entries 8–12 in Table 1). Clearly, a reduction in the polymerization temperature significantly suppresses the β-hydride elimination transfer. Figure 1 shows the proton NMR spectrum (with insets showing the expanded region and chemical shift assignment) of sP4M1P samples prepared at various polymerization temperatures. As shown, a decrease in the polymerization temperature can drastically reduce the ratio of the production of the vinylene-capped sP4M1P polymer. This temperature effect can be rationalized by the fact that a reduction in the polymerization temperature can drastically reduce the possibility of 4M1P incorporation by the unfavorable 2,1-insertion pattern. As the vinylene end group needs to be generated from a 2,1-inserted 4M1P end unit, a lower polymerization temperature disfavors the generation of vinylene-capped sP4M1P and enhances the ratio of the generation of OH-capped sP4M1P.

### 3.2. Structural Characterization of OH-Capped sP4M1P

End-group analyses provide direct evidence of the preparation of a pure OH-capped sP4M1P, which was generated through a single chain transfer pathway. As this research aims to develop a method for the preparation of sP4M1P-based stereoregular block copolymers, the successful production of a structurally well-defined OH-capped sP4M1P sample, which acts as the key prepolymer for the construction of sP4M1P-based stereoregular block copolymers, is required. Figure 2 shows the ^1^H NMR spectrum (with insets showing the expanded region and chemical shift assignment) of OH-capped sP4M1P (*M*_n_ = 9890, *M*_w_/*M*_n_ = 1.65, entry 8 in Table 1). As shown, there are four major upfield proton resonances (δ = 0.9, 1.1, 1.39, and 1.65), which correspond to the methyl -CH[CH_2_CH(*C**H***_3_)_2_]-CH_2_-, methylene -CH[C***H***_2_CH(*C*H*_3_*)_2_]-C***H***_2_-, main chain methine -C***H***[CH_2_CH(*C*H*_3_*)_2_]-CH_2_-, and side chain methine -CH[CH_2_C***H***(*C*H*_3_*)_2_]-CH_2_- proton resonances, respectively. In addition, a weak downfield proton resonance at 3.53 ppm corresponds to the methylene-CH[CH_2_CH(*C*H*_3_*)_2_]-C***H***_2_-OH proton resonance situated at a neighboring hydroxy end group. The unique multiplet coupling pattern of the methylene protons (Ha, Hb) indicates that the two protons are situated in a diastereotopic environment. 

The detailed structural assignment for OH-capped sP4M1P can be further elucidated through ^13^C distortionless enhancement by polarization transfer (DEPT) 135 and ^1^H-^13^C heteronuclear multiple quantum coherence (HMQC, supporting information Appendix A) analyses. 

To further demonstrate that OH-capped sP4M1P was generated by a single chain transfer pathway, a graph of the polymer molecular weight (*M*_n_) versus monomer/TEA ([4M1P]/[TEA]) mole ratio was plotted from the experimental data (entries 8–11 in Table 1), as polymers generated under these conditions contain only the hydroxy group end-capped sP4M1P (generated by selective chain transfer to TEA). Figure 3 shows the plot of *M*_n_ versus [4M1P]/[TEA] for a syndiospecific polymerization of 4M1P at 20 °C conducted using Me_2_C(Cp)(Flu)ZrCl_2_ as the catalyst and TEA as the chain transfer agent. The linear relationship between the molecular weight of sP4M1P and the [4M1P]/[TEA] ratio indicates that the chain transfer reaction to TEA (rate constant = *k*_tr_) competes with the 4M1P chain propagation reaction. Since the degree of polymerization (X_n_) follows X_n_ = *k*_p_ [4M1P]/*k*_tr_[TEA], the chain transfer constant can be calculated (*k*_tr_/*k*_p_ = 0.08) for the generation of OH-capped sP4M1P by the predominant chain transfer reaction to TEA.

### 3.3. Preparation of Syndiotactic 4-Methyl-1-pentene-Based Stereoregular Diblock Copolymers

As revealed by the detailed structural analyses (*vide supra*), conducting the syndiospecific polymerization of 4M1P using Me_2_C(Cp)(Flu)ZrCl_2_ as the catalyst and TEA as the chain transfer agent produces a structurally well-defined sP4M1P polymer end-capped with a hydroxy end group. Accordingly, the resulting OH-capped sP4M1P is suitable as a prepolymer for the construction of sP4M1P-based stereoregular block copolymers by using the hydroxy end group for connection onto other polymer blocks. In our study, a sP4M1P sample (*M*_n_ = 9890, *M*_w_/*M*_n_ = 1.65, entry 8 in Table 1) bearing a single hydroxy end group was allowed to fractionate into two portions to provide an end-functionalized prepolymer with a narrower range of polydispersity. The lower molecular weight portion of the OH-capped sP4M1P (*M*_n_ = 10,500, *M*_w_/*M*_n_ = 1.40) was used for the construction of sP4M1P-*b*-aPS, and the higher molecular weight portion of the OH-capped sP4M1P (*M*_n_ = 14,500, *M*_w_/*M*_n_ = 1.32) was used for the construction of sP4M1P-*b*-PMMA. The hydroxy end group of the OH-capped sP4M1P was converted into the α-bromoester end group via the esterification of the OH-capped sP4M1P with α-bromoisobutyryl bromide to produce α-bromoester-capped sP4M1P with good yield. Figure 4 shows the ^1^H NMR (with an inset showing chemical shift assignment) of the α-bromoester-capped sP4M1P sample prepared from the esterification of the OH-capped sP4M1P (*M*_n_ = 10,500, *M*_w_/*M*_n_ = 1.40). The production of the α-bromoester-capped sP4M1P can be elucidated through ^1^H NMR (Figure 4), ^13^C and ^13^C (DEPT 135), and ^1^H-^13^C HMQC (supporting information Appendix A) analyses. Finally, the construction of stereoregular diblock copolymers of sP4M1P-*b*-PMMA and sP4M1P-*b*-aPS was accomplished using the α-bromoester-capped sP4M1P as the macroinitiator, which was treated respectively with CuBr/1,10-phenanthroline (macroinitiator for MMA polymerization) and CuBr/1,1,4,7,10,10 hexamethyltriethylenetetramine (macroinitiator for MMA polymerization) for the in situ generation of active macroinitiators. Subsequently, chain extension with MMA and styrene from the respective macroinitiators by a controlled living atom transfer radical polymerization reaction allowed the production of the corresponding sP4M1P-*b*-PMMA and sP4M1P-*b*-aPS with good yield. The detailed synthetic processes for the preparation of the structurally well-defined sP4M1P-*b*-PMMA and sP4M1P-*b*-aPS stereoregular diblock copolymers are illustrated in Figure 2.

Figure 5 shows a comparison of GPC elution curves for OH-capped sP4M1P (*M*_n_ = 14,500 g/mol, *M*_w_/*M*_n_ = 1.32) and sP4M1P-*b*-PMMA (*M*_n_ = 32,000 g/mol, *M*_w_/*M*_n_ = 1.20). Figure 6 shows the ^1^H NMR spectrum of sP4M1P-*b*-PMMA (with insets showing the expanded region and chemical shift assignment). The structural characterization of the sP4M1P-*b*-PMMA sample through ^13^C (DEPT 135) and ^1^H-^13^C HMQC NMR spectra can be found in the supporting information (Appendix A).

Figure 7 shows a comparison of GPC elution curves for OH-capped sP4M1P (*M*_n_ = 10,500 g/mol, *M*_w_/*M*_n_ = 1.40) and sP4M1P-*b*-PS (*M*_n_ = 19,000 g/mol, *M*_w_/*M*_n_ = 1.34). Figure 8 shows the ^1^H NMR spectrum of sP4M1P-*b*-PS (with insets showing the expanded region and chemical shift assignment). The detailed structural characterization of sP4M1P-*b*-PS through ^13^C (DEPT 135) and ^1^H-^13^C HMQC NMR spectra can be found in the supporting information (Appendix A).

### 3.4. Self-Assembly of sP4MP-Based Stereoregular Block Copolymers

DSC thermogram of sP4M1P shows a single T_g_ at 60.1 °C (see the first heating thermogram of Appendix A) because of the homogeneous amorphous phase of sP4M1P and PMMA. After erasing the crystallization of the sPMP, two T_g_s including T_g,sPMP_ at 39 °C and T_g,PMMA_ at 106 °C can be observed, revealed by the second heating thermogram of Appendix A. This indicates the formation of the microphase-separated structure, consistent with the lamellar structure of the SAXS profile in Figure 9 (*vide infra*). The self-assembled nanostructures of sP4M1P-*b*-PMMA are investigated through SAXS measurements. The results are illustrated in Figure 9. After solution casting from THF, the SAXS profile of the sP4M1P-*b*-PMMA block copolymer exhibits broad reflections at the q values of 0.19 and 0.4. The two peaks also possess a ratio of 1:2, indicating the formation of microphase separation driven by solvent-evaporation-induced crystallization. As estimated, the crystalline thickness of the solvent-evaporation-induced crystallization is 33.1 nm. As shown in Appendix A, the melting point of the sP4M1P in the sP4M1P-*b*-PMMA block copolymer is 157.2 °C. To explore the intrinsic microphase separation without the effect of the crystallization, the sP4M1P-*b*-PMMA block copolymer is heated up to 280 °C to eliminate the sP4M1P crystallization. The corresponding SAXS profile shows multiple sharp reflective peaks with a relative q ratio of 1:2:3, indicating the well-ordered microphase separation. According to the primary reflection, the periodicity of the lamellar morphology is 41.9 nm. After non-isothermal crystallization from the melt (the fast crystallization occurs at 103.8 °C in Appendix A), the SAXS profile featuring a ratio of 1:2 indicates the preservation of the microphase-separated lamellar morphology. As a result of the crystallization-induced expansion, the lamellar periodicity increases to 57.1 nm.

In Appendix A, the melting point of the sP4M1P in the sP4M1P-*b*-aPS block copolymer is 178.6 °C. In Figure 10, multiple strong scattering peaks can be observed at 190 °C, indicating the well-defined lamellar morphology in the amorphous sample. The periodicity of the lamellar morphology is calculated as 31.5 nm. After non-isothermal crystallization from the melt (the fast crystallization occurs at 156.7 °C in Appendix A), the SAXS profile still reveals a lamellar reflection of 1:2:3. As estimated by the primary peak, the periodicity of the crystalline lamellar nanostructure is increased to 43.3 nm. As a result, the multiple reflections in the SAXS profiles verify the successful synthesis of the block copolymers. The well-defined microphase-separated lamellar morphology after crystallization suggests strong microphase separation in the sP4M1P-based stereoregular block copolymers, which is due to the enhanced persistence length by stereoregularity.

## 4. Conclusions

Our results demonstrate that OH-capped sP4M1P can be prepared by inducing a selective chain transfer reaction to TEA during the syndiospecific polymerization of 4M1P in the presence of the Me_2_C(Cp)(Flu)ZrCl_2_/MAO catalyst. The resulting end-functionalized stereoregular polymer contains a reactive hydroxy end group and hence can be treated with α-bromoisobutyryl bromide for the generation of α-bromoester-capped sP4M1P. Consequently, the in situ generation of the active macroinitiator by adding CuBr allows the chain extension of the sP4M1P block with MMA and styrene to produce the respective sP4M1P-*b*-PMMA and sP4M1P-*b*-PS stereoregular diblock copolymers. The sP4M1P-*b*-PMMA and sP4M1P-*b*-PS samples prepared using this unique synthetic process have a well-defined microstructure (with defined block lengths and narrow molecular weight distribution) and, hence, can self-organize into ordered nanostructures, as evidenced by SAXS measurements.

## Data Availability

Data sharing not applicable.

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
