# Peer review of "Syndiotactic Poly(4-methyl-1-pentene)-Based Stereoregular Diblock Copolymers: Synthesis and Self-Assembly Studies"

_polymers, 2022, doi:10.3390/polym14224815_

Round 1

Reviewer 1 Report

In this manuscript by Sung et al. a syndiotactic polymers have been synthesised by an exquisite catalyst. There are many Zr of Hf based catalyst for the same purpose. Although, I believe the results of this manuscript could be accepted for publication after following revisions.

Line 37, page 1: This sentence is not justifiable in many senses.. There are a plenty of other examples in recent times where syndiotactic polymers have been prepared with regioregularity.. need to cite them and then please rephrase the Introduction part of this manuscript. for example,

Angewandte chemie:https://doi.org/10.1002/anie.202211992

JACS: https://doi.org/10.1021/ja029780x

Line 220 page 6: What is the benefit of this typical catalyst? For purity of the catalyst free polymers, I would suggest the authors to present either CD spectra/ Polarimeter data for the new polymers?

 Finally, a better comparative discussion on the Tg could be drawn.

Author Response

Q1: Line 37, page 1: This sentence is not justifiable in many senses.. There are a plenty of other examples in recent times where syndiotactic polymers have been prepared with regioregularity.. need to cite them and then please rephrase the Introduction part of this manuscript. for example,

Angewandte chemie:https://doi.org/10.1002/anie.202211992

JACS: https://doi.org/10.1021/ja029780x

Response 1: Thanks for the valuable comments. We have rewritten line 37, page 1. The recent papers (Angew. Chem., Int. Ed. 2022, 61, e202211992. and J. Am. Chem. Soc. 2003, 125, 9062-9069.) reported by Sita et al. have been cited as suggested by reviewer. Thanks again for the helpful information.

Q2: Line 220 page 6: What is the benefit of this typical catalyst? For purity of the catalyst free polymers, I would suggest the authors to present either CD spectra/ Polarimeter data for the new polymers?

Response 2: Thanks for the valuable comments. In fact, the catalyst [Me2C(Cp)(Flu)ZrCl2] used in this study is a known catalyst (see references 25,26 of this manuscript). Thereby, purity of the catalyst can be determined using the spectroscopy data (e.g., proton NMR spectroscopy) reported in the literature. Despite the reviewer suggest to provide CD spectra or polarimeter data of the block copolymer. We need to point out that the polymer has stereoregulaity (syndiotacticity) but do not has chirality. Thereby, CD spectra and polarimeter examination are not necessary as the syndiotactic polymers and block copolymers are achiral.

Q3: A better comparative discussion on the Tg of the block copolymer could be drawn.

Response 3: Thanks for the helpful comments. Experimental results of DSC heating thermograms (for block copolymer sP4M1P-b-PMMA), which reveal the Tg value before self-assembly and Tg value after self-assembly, has been added in the discussion section (see line 349, page 12). Tanks again for the suggestion.

Reviewer 2 Report

The new methodology can be a convenient way to synthesize new generations of polymers, available to form nano-structures by self-assembly, with increased interest for the chemical, pharmaceutical industries, etc.

The present investigation offers a convenient route for the preparation of sP4M1P-based stereoregular diblock copolymers, presenting well-defined architectures and capacity to form molecular self-assembly into ordered nano-structures.
The subject is relevant for research in the field of polymers, because it brings valuable information about new methods for the synthesis of diblock copolymers, the important class of excipients usable in different industries, including the pharmaceutical field.
The authors prepared an OH-capped sP4M1P by inducing a selective chain transfer reaction to TEA during the syndiospecific polymerization of 4M1P in the presence of the Me2C(Cp)(Flu)ZrCl2/MAO catalyst and the new methodology can be considered a convenient way to synthesize new generations of diblock polymers with different applications.
Considering the major interest for biomedical applications of materials that show the ability to self-assemble with the formation of nano-aggregates, it would be interesting to perform some toxicity studies, and the preparation methodology could be oriented towards obtaining biocompatible materials.
The formulated conclusions are in accordance with the obtained results, based on the proposed study objectives.
The references are recent and relevant to the information presented.
The tables and figures are relevant and present the data accurately, being easy to interpret and understand them.

Author Response

Thanks for the helpful comments. P4M1P was commercially produced by Mitsui Chemicals at Japan in 1975 and has complied with the FDA regulations and EU regulations for use as food packaging and microwave material. P4M1P has been found to offer high gas permeability due to its extremely low density comparing with other polyolefins. As a result, P4M1P has been used as the key material for the construction of biohybrid lung as reported recently by Wiegmann (Journal of the mechanical behavior of biomedical materials, 2016, 60, 301-311.) and Liu (Journal of Membrane Science, 2022649, 120359.). Thanks reviewer for providing valuable suggestion. We agree that performing a toxicity test on the P4M1P-based block copolymers prepared in this study is important for examining their potential biological/medical applications. However our lab is unable to conduct the toxicity test by the regulation of Food and Drug Administration of Taiwan.
